Recent deterioration of coral reefs in the South China Sea due to multiple disturbances

http://orcid.org/0000-0003-3511-8933 Xiao Jiaguang
Wang Wei
Wang Xiaolei
http://orcid.org/0000-0001-6213-5208 Tian Peng
Niu Wentao wentaoniu@tio.org.cn
Third Institute of Oceanography, Ministry of Natural Resources , Xiamen, Fujian , China
Reimer James
Electronic publication date: 2022 Jul 25
Publication date: 2022
Volume: 10
Electronic Location ID: e13634
Received 2021 Jun 17; Accepted 2022 Jun 3
Copyright: © 2022 Xiao et al.
Copyright year: 2022
Copyright holder: Xiao et al.
License: This is an open access article distributed under the terms of the Creative Commons Attribution License, which permits unrestricted use, distribution, reproduction and adaptation in any medium and for any purpose provided that it is properly attributed. For attribution, the original author(s), title, publication source (PeerJ) and either DOI or URL of the article must be cited.
License URL: https://creativecommons.org/licenses/by/4.0/

Keywords: Deterioration Index, Mortality, Recruitment, Coral bleaching, CoTS, South China Sea

Funding: National Natural Science Foundation of China grant numbers 42006098, 42006128, 42006085 Scientific Research Foundation of Third Institute of Oceanography Ministry of Natural Resources grant numbers 2019003 and 2020006 This work was supported by the National Natural Science Foundation of China (grant numbers 42006098, 42006128, 42006085), and the Scientific Research Foundation of Third Institute of Oceanography, Ministry of Natural Resources (grant numbers 2019003 and 2020006). The funders had no role in study design, data collection and analysis, decision to publish, or preparation of the manuscript.

==============================
More frequent global warming events, biological disasters, and anthropogenic activities have caused extensive damage to coral reefs around the world. Coral reefs in the Xisha Islands (also known as the Paracel Islands) have been damaged following rounds of heatwaves and crown-of-thorns starfish (CoTS) outbreaks over recent decades. Based on a comprehensive community survey in 2020, we determined a diagnosis for the present state of six coral regions in the Xisha Islands. The findings suggested that these regions had a total of 213 species of scleractinian corals belonging to 43 genera and 16 families. Living coral coverage across sites was widely divergent and ranged from 0.40% (IQR: 7.74–0.27%) in Panshi Yu to 38.20% (IQR: 43.00–35.90%) in Bei Jiao. Coral bleaching prevalence was 23.90% (IQR: 41.60–13.30%) overall and topped out at 49.30% (IQR: 50.60–48.10%) in Bei Jiao. Five of the coral regions (all but Yongxing Dao) were under threat of CoTS outbreaks. High mortality combined with excellent recruitment rates suggested potential rehabilitation after recent deterioration. We employed a quantifiable Deterioration Index (DI) to evaluate the intensity of deterioration of coral reefs in the Xisha Islands. The results showed that Yongxing Dao and Langhua Jiao had low recent deterioration (DIrecent = 0.05, IQR: 0.07–0.02 and 0.04, IQR: 0.11–0.01, respectively), while Bei Jiao, Yongle Atoll, Yuzhuo Jiao, and Panshi Yu had high recent deterioration (DIrecent > 0.16). Different monitoring sites within the same coral region were heterogeneous with regards to all above indexes. Moreover, we reviewed and discussed potential disturbances that threaten the health of the Xisha Islands’ corals. It is crucial to identify severely afflicted areas and find successful methods to better manage coral reef health in this region.

Introduction

Coral reefs worldwide are suffering extensive deterioration as a result of synergic factors including natural catastrophes and anthropogenic disturbances (Halpern et al., 2008; Selig, Casey & Bruno, 2010). Reefs in the Caribbean Sea, Indian Ocean, and the Pacific region are threatened, and even remote reefs and atolls have not completely been spared (Burke et al., 2011). This deterioration can lead to the widespread collapse of healthy coral populations, losses of species richness and coral cover (Bruno & Selig, 2007), increases in macroalgae (Aronson & Precht, 2006), outbreaks of coral bleaching and disease (Donner et al., 2005), and failure to recover from natural disturbances (Edwards & Gomez, 2007). Coral reefs are losing their immense biodiversity and ecosystem functions, negatively affecting the livelihood and ecosystem services of millions of coastal people (Hughes et al., 2017).

Numerous studies have explored the drivers of coral declines and the links between climate change, human activities, and coral reef ecosystems (Hughes et al., 2003; Alvarez-Filip et al., 2011; Jackson et al., 2014; Hoegh-Guldberg et al., 2017). It is widely recognized that global threats such as ocean warming and acidification are more powerful and destructive than local threats (Gattuso, Hoegh-Guldberg & Pörtner, 2014). The prime factor, heatwaves caused by global warming, have bleached corals at increasing rates since the 1980s (Morrison et al., 2019). The State of the Global Climate announced that 2020 was one of the three warmest years on record, during which more than 80% of the ocean experienced marine heatwaves (Kennedy et al., 2021). This caused significant impacts to tropical reefs and even the subtropical fringe reefs. For example, One Tree Island, a potential refuge at the southern part of the Great Barrier Reef (GBR), had corals that were identified as severely bleached in 2020 (Nolan et al., 2021). Ocean acidification is also projected to impact all areas of the ocean with wide-ranging impacts on corals by reducing their growth rates and ability to maintain physical structure (Hoegh-Guldberg et al., 2007; Kroeker et al., 2013). A strong decrease in aragonite saturation state caused by ocean acidification was observed throughout the Greater Caribbean Region (GCR) from 1996 to 2006 (Gledhill et al., 2008). Additionally, sea level rise, hurricanes, and tropical storms are all potentially deleterious to coral reefs (Yates & Moyer, 2010; Yang et al., 2015). Among a variety of local threats, Acanthaster crown-of-thorns starfish (CoTS) outbreaks are the major contributor to sustained coral loss and degradation in many Pacific regions (Kayal et al., 2012; Baird et al., 2013). On the remote Moorea Island of French Polynesia, for example, high densities of CoTS caused severe coral loss, with more than 96% of living corals killed between 2005 and 2010 (Kayal et al., 2012). Moreover, there is a growing number of maladies that are threatening the health of corals (Woodley et al., 2016). More than 40 coral diseases have been reported, and more than 200 species of reef-building corals are affected by these diseases (Bruckner, 2009). The Caribbean Sea is a hotspot for coral diseases, and over 66% of the world’s coral diseases occur within this region (Green & Bruckner, 2000). In the Anthropocene, diverse anthropic stresses including destructive fishing and overfishing, coastal engineering, tourism industry, marine pollution, and eutrophication have aggravated the deterioration (Hughes et al., 2017). Even remote coral reefs located in the Maldives, Chagos Archipelago, Seychelles, Micronesia, and Marshall Islands are threatened by human activities (Burke et al., 2011), and stressors are projected to intensify in coming decades. Due to the above threats as well as interactions among them, the world’s coral reefs are projected to be severely compromised by 2070 (Ateweberhan et al., 2013; Morrison et al., 2019).

The Xisha Islands (also known as the Paracel Islands), one of the four groups of islands in the South China Sea, have abundant oceanic coral reefs (Huang et al., 2011). The Status of Coral Reefs of China, 2019 reported a totally of 251 species of scleractinian corals in the Xisha Islands (Huang, Chen & Huang, 2021). These islands support an immeasurable amount of marine biological resources and ecological services. However, continual disturbances have caused heavy losses to coral reef coverage and structural diversity in the Xisha Islands over past decades. In Bei Jiao, the illegal excavation of giant clams by poachers caused a loss of 13.3% in coral coverage across 2 years (Li et al., 2015). Li et al. (2018) analyzed data from 2007 to 2016 that indicated dramatic decreases in both species number and coverage due to construction activities, CoTS outbreaks, and coral diseases in Yongxing Dao and Qilian Yu. Coral communities in the Yongle Atoll were in relatively healthy conditions despite destructive fishing and overfishing practices (Zhao et al., 2016). Using survey data of 2006, Huang et al. (2011) determined that Huaguang Jiao had the highest biodiversity. By contrast, reefs affected by human activities had lower species richness, and independent reefs, such as Zhongjian Dao and Panshi Yu, were moderately biodiverse (Huang et al., 2011). Corals in Dong Dao suffered from atramentous necrosis disease in 2015 (Huang, Chen & Huang, 2021). In addition to the disturbances described above, high temperature stress and CoTS outbreaks have led to an increase in coral bleaching and mortality events in the Xisha Islands over recent years. For example, heavy ocean warming in the Xisha Islands during the summers of 2014 and 2019 caused mass coral bleaching in Bei Jiao and Yagong Dao (Li et al., 2016; Huang, Chen & Huang, 2021). Meanwhile, CoTS outbreaks relapsed in 2018 in Panshi Yu, Yuzhuo Jiao, and Langhua Jiao, when the density of CoTS reached 400 individuals per hectare. In 2019, the density reached over 1,000 individuals per hectare (Li et al., 2019). Many remote coral reefs have already been degraded despite the local anthropogenic pressures were low or even absent. Evaluating the current health conditions and deteriorative intensity of coral communities in the Xisha Islands has been an urgent task.

There have been many studies assessing the health of coral reefs, and various ecological parameters have proliferated (Díaz-Pérez et al., 2016). Specific parameters such as live coral coverage, coral recruitment, coral bleaching, species richness, and diversity index provide a fundamental diagnostic methodology when exploring the current state of corals (Risk et al., 2001; Brito-Millán et al., 2019). The Deterioration Index (DI) proposed by Ben-Tzvi, Loya & Abelson (2004) takes mortality and recruitment rates into account simultaneously. It can indicate the development trend of the health status in each coral community, not just the current ecological state. DI is different from other complex parameters such as Healthy Reefs Initiative (HRI) and Coral Health Index (CHI), and it is effective for most coral reefs, especially those with insufficient data (Ben-Tzvi et al., 2011). This quantifiable indicator evaluates the intensity of deterioration in order to assess the health of different coral communities.

In this study, we used large-scale stratified surveys to diagnose coral reefs in the Xisha Islands. We hope to provide a new comprehensive baseline for the remote reefs in the Xisha Islands. Moreover, we introduced DI as a snapshot assessment of deteriorative intensity across different regions. Our results provide evidence to help determine management actions concerning potential stressors.

Materials and Methods

Study areas

The Xisha Islands are in the northwest of the South China Sea and southeast of Hainan Island within a marine area of 15°46′N–17°08′N and 111°11′E – 112°54′E (Fig. 1). They consist of Yongle and Xuande Islands, which are comprised of a total of 39 islands, seven reefs exposed at low tides, and more than eight other submerged reefs. In this study, a total of 44 coral reef monitoring sites were established around six regions (Bei Jiao, Yongxing Dao, Yongle Atoll, Yuzhuo Jiao, Panshi Yu, and Langhua Jiao; GPS coordinates in Table S1). For the accurate and comprehensive view of the health state of these coral reefs, monitoring sites were uniformly scattered in each region to represent the different habitats of the coral communities as much as possible.

Figure 1 Six regions in the Xisha Islands with 44 coral reef monitoring sites denoted by red dots.

Field activities

Field surveys took place from 31 August to 30 September, 2020. We performed three transects of different depths (5, 10, and 15 m) at each site while SCUBA diving. Transects were demarcated using tapelines and deployed parallelly to each other. Each transect (tapeline) was 50 m in length. If the actual distribution depth of corals was less than 15 m, the deepest transect was adjusted appropriately. Point Intercept Transect (PIT) video sampling was conducted following standard procedures (Hill & Wilkinson, 2004), using a 24-megapixel Canon PowerShot G1X Mark III digital camera. In brief, a SCUBA diver held the camera with the lens 0.2–0.3 m away from the tapeline at each site, and swam slowly and uniformly along the tapeline from the starting point. The camera aimed vertically downward and shot the tapeline to record the organisms and substrate below the tapeline. The recording time was at least 10 min until the end of the tapeline. Another diver then took close-up photographs of various corals under the tapeline, and collected some specimens for species identification. Ten 50 cm × 50 cm quadrats were systematically deployed within a range of 2.5 m on both sides of the transects to take pictures using a 20-megapixel Canon PowerShot G7X Mark II digital camera.

Coral health assessment

Video transects were analyzed in laboratory using point sampling techniques, i.e., freezing the video at every 10 cm interval (scale point) to quantify the substrate and organism composition (Sample data in Data S1). Starting from the “0 m” scale point, all scleractinian corals, other sessile organisms (including soft corals, sponges, and sea anemones), dead corals, bleaching corals and substrate (reef-rock, rubble, sand, or mud) at the scale points were assessed until the “50 m” scale point. There was a total of 500 scale points in a transect. The assessment elements including: a) Species identification. The scleractinian coral species (more than 2 cm) at each 10 cm scale point were interpreted. If it was difficult to identify the species in the video, the coral close-up photographs and the coral specimens were used to assist identification. Corals were identified to their lowest tractable taxonomic level (species) following taxonomic criteria (Huang, 2018; Shi, 2019; Dai & Zheng, 2020).

b) Living coral coverage. The number of all living scleractinian corals (more than 2 cm) at scale points were counted. The number divided by 500 was the living coral coverage rate (%).

c) Coral mortality. The proportion of dead scleractinian corals to the total number of living and dead scleractinian corals at the scale points (%). Recent dead corals (dead within a year) were separated out to confirm the values of recent coral mortality. More specifically, recent dead corals are those that the corallite structures are either white and still intact, or slightly eroded, but identifiable to species. Recently dead skeletons may be covered by sediment or a thin layer of turf algae. Old dead corals are those that the corallite structures are either gone or are covered over by organisms that are not easily removed (McField & Kramer, 2007; sample data in Data S1).

d) Coral bleaching. The proportion of all bleaching scleractinian corals (not dead) to the number of living scleractinian corals at the scale points (%).

e) Coral recruitment. The number of scleractinian coral recruits (less than 2 cm in diameter or height) in quadrats were counted. The number divided by the area of quadrats was the supplement of hard coral, and the unit was ind. per m2.

Moreover, the Shannon–Wiener diversity index (Shannon, 1948) and Pielou’s evenness index (Pielou, 1966) were also used to compare the assessment results of coral health among different regions.

Disruptive factors

Natural and anthropogenic disturbances were counted and analyzed. Daily Sea Surface Temperature (SST) and Degree Heating Week (DHW) data were downloaded from Unidata (UCAR Community Programs, https://www.unidata.ucar.edu) and processed in MATLAB R2019a (MathWorks Inc., Natick, MA, USA). Daily 5 km Regional Virtual Stations Product from Coral Reef Watch (NOAA Headquarters, 2016, https://coralreefwatch.noaa.gov) for the Xisha Islands was also utilized. The number of all CoTS within 1 m of each side of the transects was counted to assess their damaging effects. Moreover, diverse anthropic activities at the monitoring sites were also recorded.

Recent deterioration analysis

We used the Deterioration Index (DI) to quantify the intensity of deterioration of coral communities among different monitoring sites and regions. The cardinal principle is that when a coral community state is stable, the DI value is expected to be low, and when a community is in decline, the DI will be high (Ben-Tzvi, Loya & Abelson, 2004). In this study, we made some adjustments that differed from the original definition under the same formula:

DI=DCDC+LC/SCLC

where DC is the number of dead scleractinian corals at scale points, LC is the number of living scleractinian corals at scale points, and SC is the number of small detectable living scleractinian corals (up to 2 cm) in 10 quadrats. We calculated a DIrecent value using the number of recently dead scleractinian corals to reflect the recent development of coral reefs.

Statistical Analyses

We summarized the assessment elements using the median (with InterQuartile Range, IQR: Q3–Q1) as opposed to the mean, by reason of the nonnormal data in the present study. Considering the small sample size in the present study, nonparametric bootstrap F-test with a pooled resampling method were applied (Dwivedi, Mallawaarachchi & Alvarado, 2017). The bootstrap sample was the same size as the original dataset and was built using sampling with replacement. This process was repeated with 1,000 replicates. Statistical analyses were performed in R 4.0.3 (R Core Team, 2020) using the “Nonparametric bootstrap F-test for comparison of three means” appendix toolbox (Dwivedi, Mallawaarachchi & Alvarado, 2017), combined with self-written scripts (R-codes in Data S2).

Results

Community health

A total of 213 scleractinian coral species (belonging to 43 genera and 16 families) were identified in the Xisha Islands. However, species richness was extremely uneven across different monitoring sites and regions (Fig. 2, Table 1). Bei Jiao presented the largest number of coral species, followed by Yongle Atoll, and Panshi Yu contributed the least. The Shannon–Wiener diversity index showed similar results: Bei Jiao had high species diversity, whereas Panshi Yu had low species diversity. Pielou’s evenness index results were generally high, ranging from 0.79 (IQR: 0.85–0.78 at Bei Jiao and IQR: 0.85–0.74 at Panshi Yu) to 0.86 (IQR: 0.86–0.82 at Yongxing Dao, Table 1). Living coral coverage differed greatly across 42 study sites, ranging from 0.2% to 49.73% (Fig. 2). No live coral was recorded at the survey site SY2 in the Yongle Atoll and the survey site PS7 in Panshi Yu. There were significant differences in coral cover across the six regions (nonparametric bootstrap F-test, p = 0.001). The characteristics of species richness and living coral coverage were approximately identical. Species richness and living coral coverage rates of monitoring sites at Bei Jiao, Yongxing Dao and Yongle Atoll (excepting the survey site SY2) were distributed more evenly than other regions, and northeast was higher than southwest area at Yuzhuo Jiao and Langhua Jiao. The species richness and living coral coverage of eastern Panshi Yu were higher than those of other parts of Panshi Yu (Fig. 2). Coral mortalities were very high, especially in Yuzhuo Jiao (42.00%, IQR: 57.80–5.85%), Panshi Yu (34.90%, IQR: 60.10–25.70%), and Yongle Atoll (15.80%, IQR: 20.90–11.50%). Recent coral mortality ranged from 0 to 33.61% (Fig. 2) with a large regional difference (nonparametric bootstrap F-test, p = 0.015). In detail, coral mortalities of monitoring sites at Yongle Atoll (excepting the survey site SY2) and PanshiYu (excepting the survey site PS7) were distributed evenly, and the south was higher than north at Yongxing Dao, Yuzhuo Jiao and Langhua Jiao. The coral mortality of northeastern Bei Jiao was higher than that of other parts of Bei Jiao. Recent coral mortalities of monitoring sites at Yongxing Dao, Yongle Atoll, PanshiYu and Langhua Jiao were more even than those of Bei Jiao and Yuzhuo Jiao. Recent coral mortalities of monitoring sites at Bei Jiao and Yuzhuo Jiao showed the same trends as coral mortalities (Fig. 2). According to our survey, corals in the Xisha Islands suffered severe bleaching in 2020. The bleaching rate was 23.90% (IQR: 41.60–13.30%) overall and topped out at 49.30% (IQR: 50.60–48.10%) in Bei Jiao (Table 1). The six regions had very different coral bleaching rates (nonparametric bootstrap F-test, p = 0.000). Coral bleaching prevalences had no obvious variation among different monitoring sites within a specific region (Fig. 2). Coral recruitment in the Xisha Islands was abundant (median 6.67 ind. per m2, IQR: 8.60–4.27 ind. per m2, Table 1), and significant differences existed across different regions (nonparametric bootstrap F-test, p = 0.012). Coral recruitment was distributed evenly among different monitoring sites within a specific region (Fig. 2).

Figure 2 Dimensional difference of health indicators across different monitoring sites in the Xisha Islands.

Table 1 Health indicators for coral communities of six regions in the Xisha Islands.

Indicators	Bei Jiao	Yongxing Dao	Yongle Atoll	Yuzhuo Jiao	Panshi Yu	Langhua Jiao	
Species richness	163	95	140	96	38	102	
Shannon–Wiener index	3.31 (3.42–3.29)	3.39 (3.50–3.20)	3.25 (3.45–3.16)	2.81 (3.07–2.73)	1.04 (1.94–0)	2.89 (3.03–1.81)	
Pielou’s index	0.79 (0.85–0.78)	0.86 (0.86–0.82)	0.80 (0.83–0.79)	0.83 (0.91–0.75)	0.79 (0.85–0.74)	0.84 (0.89–0.80)	
Living coral coverage (%)	38.20 (43.00–35.90)	19.00 (19.50–18.00)	25.50 (32.50–16.50)	11.20 (29.70–4.00)	0.40 (7.74–0.27)	12.70 (19.50–1.05)	
Coral mortality (%)	3.57 (10.90–2.44)	1.24 (10.20–0.62)	15.80 (20.90–11.50)	42.00 (57.80–5.85)	34.90 (60.10–25.70)	5.18 (7.84–3.66)	
Recent coral mortality (%)	3.57 (10.90–1.26)	1.24 (1.44–0.62)	10.20 (17.00–8.10)	14.00 (27.40–5.85)	19.30 (20.90–14.70)	2.66 (3.69–1.44)	
Coral bleaching (%)	49.30 (50.60–48.10)	13.60 (16.20–10.10)	40.10 (45.60–35.20)	23.70 (30.00–17.30)	9.88 (13.10–0)	15.00 (20.40–9.88)	
Coral recruitment (ind. per m2);	6.27 (7.00–4.93)	6.67 (8.80–5.33)	8.53 (10.50–7.20)	6.47 (9.10–5.70)	3.20 (3.80–1.83)	7.80 (9.63–6.87)	
CoTS density (ind. per 100 m2);	0.17 (0.58–0)	0 (0–0)	0 (0.67–0)	0.50 (1.58–0)	0.33 (0.33–0)	0.33 (0.67–0.25)	
DI	0.48 (1.75–0.24)	0.05 (0.57–0.02)	0.79 (2.09–0.70)	0.61 (1.01–0.28)	0.28 (1.68–0.09)	0.13 (0.21–0.04)	
DIrecent	0.48 (1.75–0.13)	0.05 (0.07–0.02)	0.50 (0.71–0.37)	0.32 (0.55–0.21)	0.17 (0.71–0.04)	0.04 (0.11–0.01)	
Note:

Species richness is represented by the total number of scleractinian coral species in each region, and other indicators are represented by the medians (IQR: Q3–Q1).

Recent deterioration

The recent deterioration of coral communities in the Xisha Islands was estimated using DIrecent values. We found noticeable differences across different monitoring sites and regions (Fig. 3, Table 1). DIrecent values ranged from 0 to 6.60 across 43 survey sites (the survey site SY2 in the Yongle Atoll was not included because there were no living corals or coral recruits). There were only six survey sites with DIrecent values greater than 1: two in Bei Jiao, two in the Yongle Atoll, one in Yuzhuo Jiao and Panshi Yu, respectively. The survey site BJ8 had the largest DIrecent value (6.60). Our results showed that Yongxing Dao and Langhua Jiao had low recent deterioration (DIrecent = 0.05, IQR: 0.07–0.02 and 0.04, IQR: 0.11–0.01, respectively), while Bei Jiao, Yongle Atoll, Yuzhuo Jiao, and Panshi Yu had high recent deterioration (Table 1).

Figure 3 Deterioration Index recent values (DIrecent) of 43 survey sites.

Inset map shows the location of different monitoring sites and regions with numbers and colors.

Multiple disturbances

Multiple external disturbances, including natural and anthropogenic triggers, were detected in the Xisha Islands in 2020, the most severe of these being rapidly increasing sea temperature. Mean SST indicated that coral reefs in the Xisha Islands suffered serious heat stress from June to September, reaching an average of 30 °C (Fig. 4A). The DHW curve showed that coral reefs in the Xisha Islands fell into bleaching alert level one & two phases from the end of July to mid-October, with an aggregated duration of more than 80 days (Fig. 4B). Corals at 93% of the sites were estimated to have affected by bleaching (Fig. 2). CoTS outbreak was another important cause of the wide-ranging demise of corals and structural destruction of reefs in the Xisha Islands. During our study in 2020, a total of 163 CoTS were found across 23 monitoring sites (Fig. 5A). The survey site YY in the Yongle Atoll had the highest population density at 29.33 ind. per 100 m2. Five coral regions (all but Yongxing Dao) were under threat of CoTS outbreaks. The attacked coral tissues showed signs of bleaching or death (Figs. 5B and 5C).

Figure 4 Sea surface temperature (SST) by month (A) and degree heating week (DHW) time series graph (B) of Xisha Islands.

MMM SST: maximum monthly mean SST, threshold SST: MMM SST + 1 °C. No stress: SST ≤ MMM SST; bleaching watch: MMM SST < SST < threshold SST; bleaching warning: SST ≥ threshold SST and 0 < DHW < 4, possible bleaching; alert level 1: SST ≥ threshold SST and 4 ≤ DHW < 8, significant bleaching likely; alert level 2: SST ≥ threshold SST and DHW ≥ 8, severe bleaching and significant mortality likely.

Figure 5 Crown-of-thorns starfish (CoTS) of Xisha Islands.

(A) Spatial distribution and density (ind. per 100 m2) of crown-of-thorns starfish (CoTS) across six coral regions. (B and C) CoTS are attacking corals, arrows point the bleaching or dead tissues.

Among several types of human activities, dynamite fishing caused the most damage to coral reefs. In this study, we discovered dynamite fishing activities in Yuzhuo Jiao and Panshi Yu that may have increased mortality at these two regions (Table 1). Moreover, tourists dived in Bei Jiao, Yongle Atoll, Yuzhuo Jiao, and Panshi Yu during our surveys. Marine litter appeared occasionally on the seabed at some of the monitoring sites.

Discussion

Given their roles in supporting marine ecosystems, biological diversity, and value to human society, coral reef ecosystems are often a center of attention. In the past 10 years, there have been many studies on the biodiversity and health conditions of the remote atolls in the Xisha Islands (Huang et al., 2011; Yu, 2012; Zhao et al., 2016; Li et al., 2019). Based on historical data and field investigations, a total of 204 scleractinian coral species were reported in the Xisha Islands in 2006 (Huang et al., 2011). In this study, we identified a total of 213 species across 43 survey sites and six coral regions. A recent study by Huang, Chen & Huang (2021) determined that there were 251 species of scleractinian corals. This shows the great variety of species and abundant hermatypic coral resources in the Xisha Islands. The median living coral coverage of the Xisha Islands was 16.50% (IQR: 33.10–5.74%) in 2020, which was much lower than the coral cover of a benchmark coral reef at the GBR in 2004 (22.00%, Sweatman, Delean & Syms, 2011; Zhao et al., 2016). The living coral coverage rates in different coral reefs either increased or decreased slightly when compared to previously reported data. An investigation of only two survey sites found that the living coral coverage in Bei Jiao was 50.84% in 2014 (Li et al., 2015). In this study, the median living coral coverage of Bei Jiao was 38.20% (IQR: 43.00–35.90%), and ranged from 7% to 49.73% across eight different sites (Fig. 2). The living coral coverage in Yongxing Dao declined sharply from 46.67% to 5.00% from 2007 to 2016 (Li et al., 2018). However, our data showed that the living coral coverage in Yongxing Dao was 19.00% (IQR: 19.50–18.00%) in 2020 (Table 1). The coral cover in the Yongle Atoll was 25.50% (IQR: 32.50–16.50%) in 2020, which was also better than the coverage in 2013 (18.00%, Zhao et al., 2016). Moreover, a good level of coral recruitment (median 6.67 ind. per m2) also indicated the relative “health” of corals in the Xisha Islands (Healthy Reefs Initiative, 2008). In fact, coral recruitment in the Xisha Islands increased annually after 2015, reaching 3 ind. per m2 in 2019, and some islands such as Lingyang Jiao and Jinqing Dao in the Yongle Atoll reached 5 or 6 ind. per m2 (Li et al., 2019). In 2020, Yongle Atoll’s coral recruitment was 8.53 ind. per m2 (IQR: 10.50–7.20 ind. per m2), and Langhua Jiao’s reached 7.80 ind. per m2 (IQR: 9.63–6.87 ind. per m2) (Table 1). These indicators mentioned above seem to indicate that the coral communities in the Xisha Islands were in a relatively healthy state, although the truth is that the coral reefs in the Xisha Islands have been suffering extensive deterioration over recent years when we take Deterioration Index (DI) values into account.

Coral reefs in the Xisha Islands have been under pressure due to complex reasons but currently, the most urgent challenges are rising temperatures and coral predators (Wu et al., 2011; Li et al., 2016; Li et al., 2019). Since 1998, the frequency, intensity, and duration of heat stress events have worsened as global warming increased, thereby increasing the impact of these events on coral reefs and other marine systems around the world (Heron, Eakin & Douvere, 2017; Hughes et al., 2018; Eakin, Sweatman & Brainard, 2019; Morrison et al., 2019). The most recent mass coral bleaching event from 2016 to 2017 caused unprecedented damage to nearly all coral reefs. In Australia, studies have shown that about 93% of the GBR was bleached (Heron, Eakin & Douvere, 2017). Reefs in the Chagos Archipelago, central Indian Ocean, suffered severe bleaching and mortality, and their coral cover decreased from 30% to 12% in 2016 (Head et al., 2019). The Maldives experienced major bleaching, with 73% of corals bleached across 71 survey sites (Ibrahim et al., 2017). The Pacific Island nations of Palau and the Federated States of Micronesia were also ravaged by this mass coral bleaching event (NOAA Headquarters, 2016). Even Sesoko Island, Okinawa, a high latitude region, saw the bleaching of 99.2% of its colonies in 2016 (Sakai, Singh & Iguchi, 2019). Coral bleaching events have also been recorded in the Xisha Islands in 2014 and 2019 (Li et al., 2016; Huang, Chen & Huang, 2021). 2020 was announced as one of the three warmest years on record, signifying a new round of mass coral bleaching. Anomalous temperatures at the beginning of 2020 caused widespread bleaching across the GBR, extending to those previously less-affected reefs such as in One Tree Island, and almost half of the surveyed live hard coral cover was bleached (Nolan et al., 2021). Similarly, in 2020, the worst known coral bleaching affected the Xisha Islands, with an median of 23.90% bleaching prevalence (Table 1). The mean SST of the Xisha Islands from June to September 2020 was 30.12 °C, which was 1.05 °C above the mean value of the region’s previous record for the same period from 1990 to 2020. Moreover, IPCC-RCP4.5 forecasted that the global-mean temperature will increase 2.4 °C by 2,100, which exceeds the level of warming (1.5 °C) that can induce severe degradation of a great majority of coral reefs (Frieler et al., 2013; Schleussner et al., 2016). Against the backdrop of global warming, coral bleaching in the Xisha Islands may become a normal occurrence in the future.

Additionally, the Xisha Islands are now in the middle of their second CoTS outbreak. During the first outbreak of 2007–2009, the mean density of CoTS reached 255 ind. per 100 m2 in the ecological monitoring area (Wu et al., 2011). Since 2018, a new CoTS outbreak has developed. In Panshi Yu, Yuzhuo Jiao and Langhua Jiao, the mean density of CoTS was 4 ind. per 100 m2 in 2018 and increased to 10 ind. per 100 m2 in 2019 (Li et al., 2019). In 2020, although the density of CoTS was 0.33 ind. per 100 m2 (IQR: 0.67–0 ind. per 100 m2) across 43 survey sites, the survey site YY in the Yongle Atoll reached a staggering density of 29.33 ind. per 100 m2 (Fig. 5A). This was far beyond the tolerable limit for a healthy coral reef (0.15 ind. per 100 m2, Moran & De’ath, 1992). Li et al. (2019) showed that the cycle of CoTS outbreaks in the Xisha Islands was about 15 years, consisting of a 5-years outbreak period and a 10-years recovery period. Therefore, CoTS in the Xisha Islands will be still in a high-density status over the next 2 years. Climate change and the decline of natural enemies have accelerated the outbreak period, and CoTS have become a time bomb threatening the health of Xisha Islands’ corals. CoTS, the largest and most destructive predator of scleractinian corals, have broken out four times since the 1960s (Pratchett et al., 2017). They are the main contributor to sustained declines in coral cover and the degradation of coral reefs at many locations throughout the Indo-West Pacific, such as Australia, Japan, Philippines, French Polynesia, and some island nations in the Indian Ocean (Trapon, Pratchett & Penin, 2011; De’ath et al., 2012). In the GBR, one-third of coral reef damage has been attributed to CoTS predation (Timmers et al., 2012), and in the Ryukyu Archipelago, at least two rounds of CoTS outbreaks have decimated corals (Nakamura et al., 2014). Unfortunately, despite more than 30 years of effort, there is neither a clear understanding of the initiation and spread of outbreaks, nor an effective means of intervention (Pratchett et al., 2017).

An improved DI method was employed for the first time to quantify the deteriorative intensity of coral reefs across different regions in the Xisha Islands. Our results showed that DIs in Bei Jiao were extra high (DIrecent = 0.48, IQR: 1.75–0.13) (Table 1). Li et al. (2015) found that a giant clam excavation in 2012 caused destruction and loss of corals in Bei Jiao. Living coral coverage fell by 13.3% and coral mortality rose over 2 years. China’s government has banned the excavation and transaction of giant clams and their processed products since January 1, 2017. According to our survey results, the coral mortality within the period of 1 year in Bei Jiao was clearly still too high (3.57%, IQR: 10.90–1.26%), and we believe global warming to be largely accountable for this, along with CoTS and dynamite fishing. Yongxing Dao, the location of Xisha administrative region, showed low deteriorative intensity of coral reefs in this study (DIrecent = 0.05, IQR: 0.07–0.02). As previously mentioned, complex factors have degraded the living coral coverage and coral recruitment in Yongxing Dao over the past decade (Li et al., 2018). However, our results indicated that coral reefs in Yongxing Dao may undergo rehabilitation following a major disturbance. The DIs in this study indicated that Yongle Atoll’s coral reefs were suffering deterioration (DIrecent = 0.50, IQR: 0.71–0.37), which is a different conclusion compared to that of Zhao et al. (2016). The coverage of dead coral was very different (0.80% in Zhao et al. (2016) vs. 10.20% in the present study). One possible cause of this could be the high-density of CoTS at some survey sites in the Yongle Atoll. The recent deterioration in Yuzhuo Jiao and Panshi Yu should also be noted (DIrecent = 0.32, IQR: 0.55–0.21 and 0.17, IQR: 0.71–0.04, respectively), which may be caused by dynamite fishing activities.

It is important to highlight the absence of long-term continuous coral reef health reports for the Xisha Islands. A single survey is laborious to estimate the trend of deterioration. A lack of baseline data is unfavorable for grading the heath states of coral reefs. The Healthy Reefs Initiative published a series of benchmarks and red flags for the Mesoamerican Reef Region (McField & Kramer, 2007). Based on this, an expected DIrecent value of the Mesoamerican Reef Region is 0.12–0.16. Using the DIrecent values, we can get a snapshot assessment of deteriorative intensity across different regions in the Xisha Islands. In any case, the long-term data collected systematically over the appropriate geographic scales is crucial to estimate the health status of coral communities in the Xisha Islands using DI. There are, of course, some disadvantages in this method, which one should be aware of prior to any attempt of applying it. DI cannot reveal changes in the community structure and sometimes the results obtained can be biased. In summary, it should be stressed that we have no pretension to present the DI as an alternative to all other reef health indices. We propose the DI as a fast and easy index, which can be applicable across diverse coral reefs.

Helping coral reefs to keep their health is a profound challenge for managers and scientists. In this context, reef governance generally includes scientific actions on ecology, economy and human society (Hughes et al., 2017). In Australia, zoning of the GBR marine reserve network appears to be making major contributions to the protection of coral biodiversity, ecosystem resilience, and social and economic values (McCook et al., 2010). In Caribbean, hundreds of marine protected area (MPAs) and long-time continuous monitoring have helped managers to make the right decisions including fisheries management strategies, simplify and standardize coral monitoring, adaptive legislation and regulations (Jackson et al., 2014). In China, at present, both the governance and management of coral reefs are typically focused at the local level and on the regulation of proximal drivers (for example, pressure from fishing or nearby coastal development). Learning from the experience of other regions is quite necessary to improve the ability to protect and manage Xisha Islands’ coral reefs. For example, it is important to raise citizens’ marine environmental consciousness and strengthen marine environment protection, reduce discharging pollutants directly into the sea, establish coral protected areas, enact strict fishing policies, and restore coral reefs. For the start of these processes, scientific assessments of coral health as in the current study are crucial for decision-making in the Xisha Islands.

Conclusions

In this study, our large-scale stratified survey revealed comprehensive diagnoses of six coral regions in the Xisha Islands. DIs showed that coral reefs in the Xisha Islands are suffering extensive deterioration as a result of natural and anthropogenic disturbances. Coral reefs of Yongxing Dao and Langhua Jiao showed low recent deterioration, while Bei Jiao, Yongle Atoll, Yuzhuo Jiao, and Panshi Yu reefs had high recent deterioration. These results highlight the need for comprehensive management actions of the coral reefs in the Xisha Islands. Moreover, continuous monitoring using DI is one component to estimate the long-term trends of coral communities.

Supplemental Information

Supplemental Information 1 Longitude and latitude information of 44 coral reef monitoring sites at six regions.

Click here for additional data file.

Supplemental Information 2 Sample pictures for bleaching corals, coral recruits, dead corals (including old dead corals and recent dead corals) and living corals.

None

Click here for additional data file.

Supplemental Information 3 R scripts for nonparametric bootstrap F-test analysis.

Click here for additional data file.

Supplemental Information 4 Raw data for analyses in the tables and figures.

Click here for additional data file.

We would like to express our gratitude to Dr. Wang Liangming and Kuang Fangfang for their assistance with temperature data processing. We also thank the reviewers for their constructive and thorough comments that improved the manuscript.

Additional Information and Declarations

Competing Interests

Author Contributions

Data Availability

The authors declare that they have no competing interests.

Jiaguang Xiao conceived and designed the experiments, performed the experiments, prepared figures and/or tables, authored or reviewed drafts of the article, and approved the final draft.

Wei Wang analyzed the data, prepared figures and/or tables, authored or reviewed drafts of the article, and approved the final draft.

Xiaolei Wang performed the experiments, analyzed the data, prepared figures and/or tables, and approved the final draft.

Peng Tian performed the experiments, analyzed the data, prepared figures and/or tables, and approved the final draft.

Wentao Niu conceived and designed the experiments, authored or reviewed drafts of the article, and approved the final draft.

The following information was supplied regarding data availability:

The raw measurements are available in the Supplemental Files.

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
