# Peer review of "Recent deterioration of coral reefs in the South China Sea due to multiple disturbances"

_PeerJ, doi:10.7717/peerj.13634_

## Round 0.1 · original submission · Major Revisions

I have heard back from two expert reviewers who are both familiar with the South China Sea. While both realize that your work has value, as data from this part of the world are scarce, there are numerous issues to address before your work can be published. I have read the paper myself and agree with the large majority of the reviewers' scientific comments. As well, the English in your manuscript needs much revision. Upon revision please supply me with the name of the person or company who proofread your paper. Upon the next submission, I will check the English before sending it out to review, and may send it back to you immediately if it still needs work.

I look forward to seeing your revised version.

·

Basic reporting

Clear and unambiguous, professional English used throughout - No

Literature references, sufficient field background/context provided - Kind of OK (see in comment to authors)

Professional article structure, figures, tables. Raw data shared - Kind of OK (see in the comments to authors)

Self-contained with relevant results to hypotheses - OK

Experimental design

Original primary research within Aims and Scope of the journal - Yes

Research question well defined, relevant & meaningful. It is stated how research fills an identified knowledge gap. - Not really (see in comments to authors)

Rigorous investigation performed to a high technical & ethical standard - Not really (see in comments to authors)

Methods described with sufficient detail & information to replicate - Kind ok OK (see in comments to authors)

Validity of the findings

All underlying data have been provided; they are robust, statistically sound, & controlled. - This part is OK

Conclusions are well stated, linked to original research question & limited to supporting results. - Not really (see the comments to authors)

Speculation is welcome, but should be identified as such. - NA

Additional comments

After reading the manuscript, I feel it is difficult to understand what authors want to say because the English is not clear due to grammatical errors and error in sentence.
This starts right from the abstract. Having said that, it is important to understand the dynamics and condition of reefs in remote atolls and islands. Hence this work should be encourage, but not before it is revised to have a more comprehensive introduction.

Introduction
The introduction is very simple and does not contain all the information necessary to make way for discussion. For example, information on the aspects of how, even remote, some atolls and islands cannot escape from natural and anthropogenic disturbances. Introduction does not have any information or comparison of other research conducted on remote atoll and island and hence fail to state why this work is important. There are information, but I feel they end and star abruptly without any hypothesis or question (there is in the last paragraph of the Introduction, but it is still lacking)

Materials and Methods
Experimental design is fine, however, authors need to give more details about the equipments used in videography including resolution
It is necessary to include the photos from which you extracted the data. Readers need to see those photos. Since authors have used the video transects to extract all the data they have mentioned it is necessary to be more clear how they did it. Did they freeze the video at every 10cm intervals to and then analyse the benthos?
Please provide representative photos for each of those indicators, especially coral recruits.

Results
Authors do not mention how did they identify coral species and to what level?
Also results have more really weird sentences - for example - Line 176,177 - "DIrecent of survey site SY2 at Yongle Atoll was illegal since no coral and recruit have been found"

Figures - Please provide map of China, which will help readers to know where in the South China Sea - Xisha is present. In its present form Figure 1 is not complete

Discussion
Needs lots of work. I cannot understand because many sentences are not properly written. Also, since the Introduction is not strong enough, the discussion suffers.

For example, Line 215-217- Authors state - "To sum up, a scene of prosperity spreads out, but when shift the
current assessment paradigm, a series of risks or impacts expose the nature of deterioration at Xisha Islands" don't underhand this. Does it mean the coral reef is healthy as of now, but due to temperature and COTS it will not be in the future? or it is not healthy at present, as per your DIrecent results and also contradicts next paragraph..

So, from the discussion, this is not the 1st record or highest record of COTS outbreak in Xisha? If so, this does not clear in the introduction, and is misleading. This study found only 163 individuals of COTS, previously it was 925?

I suggest complete revision of Introduction and Discussion such that the introduction is more detailed to include all the necessary information from previous studies in Xisha as well as comparative information. This will help to have a better discussion as well

·

Basic reporting

This is a phenomenon-reporting manuscript of the current status of coral reefs in Xisha Islands, South China Sea (also known as Paracel by western countries). Some hypotheses were raised, but no further testings.
The explanations were reasonable.
English is generally OK, although I do have doubts about a few sentences. Some help from an English editor could take care of this.
High-quality figures, although more details are needed for some figures (notes in other sections.).

Experimental design

The basic design was to choose and survey 6 reefs, each with multiple transects and quadrats.
The quadrats were deployed randomly according to the text. That is unlikely in the real world. I suggest changing “randomly” to “haphazardly” or “systematically” whichever was the real case. The number of transects/quadrats was high and there was little doubt that the pattern found should be representative of 2020 in Xisha Islands.
The South China Sea has many atolls, most are difficult to access.
It would be nice if coordinates of those transects (no need for small quadrats) were mentioned. It makes temporal comparisons much practical.

Validity of the findings

The study is based on field works. Whereas dead/alive, small/large bleached/healthy are easy to judge underwater, hundreds of coral species are not easy to distinguish from one another underwater. The authors may want to write something about this issue.
I am not sure if the authors have uploaded all raw data. I do have suggestion/doubt about the statistical methods.
The authors need to justify using one-way anova to compare those indices. The raw data for the statistical procedures were 1) indices based on occurrence/frequency data, 2) relative frequencies, mortality, 3) percentages or densities. As a replacement, Resampling method could be considered.

Additional comments

Fig 2: Units are needed for every subfigure, e.g., coral bleaching, coral recruitment…
Fig 3: There were several sample reefs within each island. One has to wonder how are those reefs/sites arranged in the figure. Is there any meaningful way adopted? They should certainly be mentioned in the legends.
Fig 4: How were data of SST processes. What I really wonder was how come the frequency distribution of temperature were mostly “normal” within month, except two?
English is fine except some sentences needs revision to be clear. For example, I had problem with the following sentences.
1.Species distribution of hermatypic corals was not balanceable among different monitoring sites and regions.
2. DIrecent of survey site SY2 at Yongle Atoll was illegal since no coral and recruit have been found.

---

## Round 0.2 · Major Revisions

I have heard back from the same two reviewers, who find your work improved, but still have many comments that you need to address. Both reviewers recognize the importance of this work, and thus I hope you can thoroughly address their concerns.

As well, please note that the English, although somewhat edited, is still in need of serious editing. I will send this paper back to you without review upon resubmission if the English is not up to international standards and easy to follow and understand. I am sorry to be strict on this, but I do not want any language barriers to impede any final judgments on the acceptance of this work. Please be thorough in this regard.

·

Basic reporting

Although the English has improved from the previous version. I still find it difficult to understand, especially methods section

For example, Line 130 - "The transects (tapelines) were totally 50 m in length" - does this mean each transect was 50 m or total of 3 transects at 3 depths amounted to 50 m?

Another example, Line 131 - "If the actual distribution depth of corals was less than 15 m, the deepest transect can be adjusted appropriately" -- authors use "can be", should be "was"?

So, please check for English usage again

Experimental design

I do have comments now - previously i did not because of not clear writing

I don't agree with authors using DI to assess the stability and degradation of coral health since the equation for this index (even after the authors modify it) is ambiguous

For example, if DC = 10, LC=2 and RC = 1 and
DC =2, LC=10 and RC = 1 the DI will be same,

however, if you add 1 more recruit to either one, the value will change by half (increase or decrease)

This is also the reason you see that even with about 36% and 24 % of live coral cover, you still find some locations with high DI

Because the equation is biased towards "recruits or small corals". Even if you miss inadvertently 1 or 2 recruits during analysis or add 1 or 2 the health can shift from stable to degrading and vice versa

In table 1 you show the recruitment to be quite stable across locations except in Panshi Yu.

SO, I don't know how will you handle over or under estimation of recruit number

Validity of the findings

no comment

Additional comments

After reading the revised version of the manuscript, the authors have revised according to the comments and suggestions from both the reviewers. I feel the information presented in the manuscript is useful for researchers in the South China Sea area as well as ecologists elsewhere.

However, i do not agree with the usage of DI since it cannot present true picture and some times DI for stable and degraded reef can be same (See above - Experimental Design section)

Removing the DI part from the manuscript will not alter the relevancy of this report, since the data presented on coral reef health assessment and other parameters including COTS is important

I suggest authors to consider removing the DI part from the manuscript and revise accordingly

·

Basic reporting

Peer J review


This investigation reports coral status in Xisha Island (Paracel Island). Basic data, e.g., bleaching percentages, recent mortality percentages were provided in quite a few locations. These islands were off-limit to tourists or foreigners; they are nevertheless in the frontline of global changes. A substantial amount of effort and resource is invested in carrying out this survey


Experimental design

Using atolls/islands as a unit is the way this study chose. The alternative is use habitats within each atoll/island since there are great difference among different habitats. This part, i.e., characteristics of surveying sites, need more details. Some significant patterns of bleaching and mortalities should emerge among habitats.

Validity of the findings

Given the snapshot nature of the investigation, and the fact that the sites are remote and protected, there could be little room to doubt the validity of the findings reported here. There is little justification for such doubt, either.

Additional comments

4. General comments
How were surveying sites determined? This could easily “determine” or bias the results as variation among sites were high. For the same reason, the averages, e.g., coverage of 19.41%, needs to be accompanied by error ranges. In fact, using “mean” does not seem to make sense in this study, since the distribution of those numbers must differ from bell-shaped, , “median” should be a better choice. These error ranges of either means or medians, estimated by bootstrapping method, may not be symmetric, but they are nevertheless necessary to determine if difference among surveys of different years and sites are statistically significant.
Xisha Islands are known as Paracel in western world. Put this in the title, e.g., in parenthesis, could increase interest and a sense of relevancy.
“Recent dead corals (dead within a year)” needs to be better defined, so others could consider adopting it. Show some contrasting pictures distinguish Recent dead and other dead corals may be helpful.
“DI is a reliable criterion and efficient tool for assessing health of coral communities, and it can provide a comparable quantitative indication for the deteriorative process and its intensity. “ I agree with the part of “quantitative”, other parts belong in Discussion.
“DIrecent better reflects the real recent coral reef development. “ Better than what? This statement does not belong in M&M, anyway.
Some statements, e.g., “More precisely, coral reefs have been in a cycle of disturbance and recovery.” seem to be based on believes without data or proper citations.
“Based on 15 years of monitoring data, Li et al. (2019) showed that the cycle of CoTS outbreaks in the Xisha Islands was about 15 years…” 15-year cycle based on 15 years of data? This sentence needs change.
What does “full-scale coral reef health reports” mean? What could it accomplish that this study could not? How could the “full-scale …reports” be implemented, and help reversing the declining coral reefs? Give some examples of how “full-scale…” data could help.
Rather than focusing on comparison within Xisha Islands, more effort should be spent on comparisons with other parts of the world. It may reveal the factors impacting the reefs in SCS or under Chinese management.
The title implies that there are linking between the decline of coral reefs and multiple factors. The authors provided some subjective opinions, whereas most are in the stage of “reasonable doubt”. It would be more convincing if the multiple factors mentioned also have a temporal trend. Which of those environmental factors played a dominant role, for example. It would be more valuable if some evidence in these directions are available.

---

## Round 0.3 · Minor Revisions

I have heard back from two reviewers, who have offered some comments. In particular, I agree with reviewer 1's insistence on a disclaimer regarding your methodology, and this should be added in a frank discussion of your results (in the Discussion).

·

Basic reporting

no comment

Experimental design

no comment

Validity of the findings

no comment

Additional comments

I commend the efforts the authors have put to revise the manuscript and also answer the comments of the reviewers.
With respect to the explanation given by the authors about the usage of DI to determine the health of the coral reefs is still not satisfactory.

While I get the point, still think that the results obtained by using the formula can be very biased against or depending on many factors - for example - what happens if another transect is laid adjacent or a few meters to the location of the transect in this work? there is every chance it will skew the results as even an addition or reduction of 1 can change the result. I did mention this and the authors say that this is a kind of extreme example. I don't think so. Even what you see on a transect laid at any location is random, the observation bias, as well as community dynamics in a given location (on a wide scale and not just that seen on the transect), will result in this method questionable

The authors mention that since they did 3 transects at each site ("In this study, we calculated the coral recruits of 10 quadrats in a transect, and in general, SC is large enough and tolerance, one more or one less recruit could not change the general situation. What's more, we did three transects (30 quadrats totally) in a survey site")...yes but this is not 3 replicate transects at each depth, but rather an independent single transect at 3 different depths. Dynamics can vary with depth as well as at each depth with distance. This is exactly why I say that the results using the equation for DI can be biased unless there are enough replicates with surveys done across and time.

I wonder why this method is not popular and used generally by coral reef ecologists if it is effective?

So, I suggest that the authors put a disclaimer and mention that this way of assessing the health has its own caveats and the readers need to be very careful when interpreting the results of the work and also pay attention if they want to use this method.

Other than this, I don't have any more comments on this manuscript

·

Basic reporting

Professional English. I marked some sentences that needs clarification.
This is a report of surveys, not a hypotheis-testing article.

Experimental design

no comment

Validity of the findings

See 1. Basic reporting

Additional comments

no

---

## Round 0.4 · Minor Revisions

Thank you for your revision; it is now scientifically acceptable for publication. The English needed a final brush-up; and I have done so here with the attached file. I am sorry, I should have mentioned this in the last round of review, but hope you can look at my edits and re-submit your work quickly. I anticipate being able to accept it at that time.

---

## Round 0.5 · accepted · Accept

Thank you for your perseverance; I am pleased to accept this work and move it into production.